# Experimental Study of Drilling Temperature, Geometrical Errors and Thermal Expansion of Drill on Hole Accuracy When Drilling CFRP/Ti Alloy Stacks

**DOI:** 10.3390/ma13143232

**Published:** 2020-07-20

**Authors:** Vitalii Kolesnyk, Jozef Peterka, Marcel Kuruc, Vladimír Šimna, Jana Moravčíková, Tomáš Vopát, Dmytro Lisovenko

**Affiliations:** 1Sumy State University, Rymskogo-Korsakova Str., 2, 40007 Sumy, Ukraine; v.kolesnik@tmvi.sumdu.edu.ua (V.K.); d.lisovenko@omdm.sumdu.edu.ua (D.L.); 2Faculty of Materials Science and Technology, Slovak University of Technology in Bratislava, Vazovova 5, 81243 Bratislava, Slovakia; marcel.kuruc@stuba.sk (M.K.); vladimir.simna@stuba.sk (V.Š.); jana.moravcikova@stuba.sk (J.M.); tomas.vopat@stuba.sk (T.V.)

**Keywords:** CFRP/Ti alloy stack, drill thermal expansion, drilling temperature, hole diameter, roundness

## Abstract

The drilling of holes in CFRP/Ti (Carbon Fiber-Reinforced Plastic/Titanium alloy) alloy stacks is one of the frequently used mechanical operations during the manufacturing of fastening assemblies in temporary civil aircraft. A combination of inhomogeneous behavior and poor machinability of CFRP/Ti alloy stacks in one short drilling brought challenges to the manufacturing community. The impact of the drilling temperature and time delay factor under various cutting conditions on hole accuracy when machining CFRP/Ti alloy stacks is poorly studied. In this paper, the drilling temperature, the phenomenon of thermal expansion of the drill tool, and hole accuracy are investigated. An experimental study was carried out using thermocouples, the coordinate measuring machine method, and finite element analysis. The results showed that the time delay factor varied from 5 (s) to 120 (s), influences the thermal-dependent properties of CFRP, and leads to an increase in hole roundness. Additionally, the thermal expansion of the drill significantly contributes to the deviation of the hole diameter in Ti alloy.

## 1. Introduction

Drilling is the most widespread machining operation when producing holes for clamping FRP/metal stacks in the aerospace industry. Though the question of ensuring hole quality in FRP/metal has been in the focus of research activity for the last 15 years, drilling CFRP/Ti alloy stacks in a single shot is still a challenging task.

Hole quality in CFRP/Ti alloy stack is characterized by the following respond values in Ti: Surface roughness [1], hole diameter, geometrical accuracy (roundness) [2], burr size [3]; and in CFRP by delamination factor (*F_d_*) [4,5], thermal destruction, and the damage value (*Q_d_*) [6]. In experimental studies the most often measured physical quantities are thrust force (*F*, N) [1,5], torque (*M_c_*, Nmm) [1,5], cutting temperature (T, °C) [7,8,9,10,11,12,13,14], chip formation mechanism [14,15], and technical quantities such as flank wear [16,17,18] and tool life [19]. The abovementioned quantities are commonly used to explain the physical nature of cutting parameters [3], the tool geometry [20,21], and tool material, as well as the coolant application effect [7,22,23,24] on the response values. In an experimental study of Ti/CFRP/Al stack drilling, it was found that the feed rate (0.05–0.15 mm/r) had a significant impact on thrust force with percentage contribution ratios (PCR) of 40% in Ti, 31% in CFRP, and 20% in Al, as well as on the torque with PCR‘s of 72% in Ti and 24% in Al [3]. Feature research identified that drilling the CFRP/Ti stack with a cutting speed *v_c_* = 45 m/min and feed *f* = 0.09 mm/r via a WC drill with a thickened chisel edge, point angle of 130°, and helix angle of 20° would provide hole a diameter according to H9 and a hole of roundness of 0.015 mm. It was specified that the most significant factors that responded to delamination in CFRP are feed rate [19,25,26] and point angle [26,27]. However, the burr size in Ti alloy is mostly affected by the feed rate [28], tool material [29], and tool wear [17,18], and it could be reduced by using MQL [7,24,29], LN_2_ [16,30,31], VAD [22] and UAD [10,23] techniques. According to Xu and Mansori [27], the surface roughness in CFRP when drilling CFRP/Ti stack is 3 times worse than when drilling only CFRP, which was explained by Titanium chip sliding in CFRP hole wall [29]. The study of Benezech et al. [20] concerning CFRP/metal stack drilling, with a point angle variation (120−150°) and rake angle (10–40°), shows that the optimal drill geometry is 135° for the point angle and 30° for the rake angle because of short chip formation which decreases the wall integrity. One more significant factor which influences the hole diameter regardless of the feed rate is the parameter of the drill’s margin [24]. It was established that the triple margin drill could produce more accurate holes due to the reduction of vibrations because of increasing contact with the machined surface [32]. On the contrary, the height of the margin could increase the radial force, cutting torque, and the friction coefficient, which leads to hole shrinkage phenomenon expressed as a variation in hole diameter and roundness [33]. Tool wear when drilling CFRP/Ti stacks is usually characterized by flank wear [32] and rare chipping of the cutting edge [34]. It was found that tool wear, as a significant circumstance of ensuring roughness in CFRP and hole size in CFRP/Ti stacks, is mostly affected by the feed rate [32]. At the same time, the spindle speed corresponds to the flank wear effect on the delamination factor [4,5] and burr size [18,22,35].

Based on the abovementioned information, it could be said that at present, there are several directions for providing research on the study of hole accuracy and quality in CFRP and CFRP/metal stacks [36]. One of them is the study of physical quantities such as thrust force, torque, and cutting temperature for a better understanding of the machining of CFRP and stacks to find ways of ensuring hole quality and accuracy. Thrust force and torque during drilling of CFRP/metal stacks are well-studied, but the cutting temperature and its effect on the hole quality in the CFRP/metal stack still are in the focus of scientific research.

Currently known methods and techniques for measuring cutting temperature can be classified into calorimetric, thermocouples [10,12,13,14,17,37,38,39], and thermographic methods [7,10,11,12,13,15,22,23,31,37]. The most common methods of measuring cutting temperatures during machining CFRP/metal stacks are the thermographic and thermocouple methods.

In a comparative study of experimental and numerical modeling of CFRP milling with different type cutters using the thermographic method, it was measured that the temperature of the cutter raised to 250 °C regardless of the cutter type, cutting speed, and feed. Using the thermocouple embedded into the CFRP workpiece, it was found that only 16% of the heat energy moved in a workpiece, from 30% to 46% in the cutter, and the rest transferred into the chip and the air [13,37]. In the experimental study, it was determined that, for avoiding delamination damage, the drilling area temperature should be lower than the glass transition temperature but higher than the brittle deformation of the epoxy resin [38].

During the milling of UD-CFRP with a fiber orientation which varies from 0° to 135°, with a 10 mm ball end mill using various cutting speeds (200–375 m/min) and feeds (0.063 mm/r), the temperature was measured in the mill through the K-type thermocouple embedded into the tool and wireless transmission system. It was found that the fiber orientation affected the cutting temperature significantly [9]. In an experimental study of the investigation of the thermal damage of the CFRP and hole size, drill temperature was measured using an infrared camera. Drill diameters vary from 4.09 mm to 12.94 mm, cutting speed ranges from 38 m/min to 112 m/min, and feed rates of 0.07 mm/r to 0.17 mm/r. It was determined that increasing drill temperature leads to an increase of hole size deviation, regardless of the high feed rate, because of poor chip removal conditions [11]. While measuring the drilling temperature with the wireless device embedded in a twist, brad spur and multi-faceted drills of the K-type thermocouples are used when drilling the CFRP/Ti alloy stack. This research was conducted under various cutting speeds in stack layers and equal feed rates. It was given that the drilling temperature decreased during the evaluation of the feed rate in the CFRP (122–85 °C) and increased in Ti alloy (182–260 °C) [14] because of the low thermal conductivity of Ti alloy; 80% of the heat generated in the cutting zone accumulated in the tool. This provokes a high drilling temperature and leads to an increase in burr height [15]. It was concluded that brad and multi-faceted drills produced lower drilling temperatures in comparison with the twist drill because of better chip evacuation conditions [14]. On the other hand, when drilling the CFRP/Al and Al/CFRP stack, it was shown that because of the excellent thermal conductivity of the Al alloy, the heat accumulation in the drill during the drilling of CFRP, as well as heat from the cutting of Al, defused rapidly and led to a decrease of the cutting temperature to 117 °C [8].

In the introduction, it was outlined that the effects of cutting parameters, tool geometry, and tool wear mechanisms on hole quality when drilling CFRP/Ti stacks are well studied. However, the drilling of CFRP/Ti alloy is provided under dry machining conditions because of the high hydroabsorption properties of CFRP. This presents researchers and engineers in the field with the problem of dissipation of the heat generated in the cutting zone, mostly impacted by the machining of the Ti alloy layer of the stack. Though the high cutting temperature is a source of such defects of machining like the thermal degradation of CFRP, delamination, and fiber pull-out, there are several studies of the cutting temperature’s effect on the quality parameters of holes. Moreover, in several papers [11,15,27], the authors reported the suspicion that the hole size could be affected by the drilling temperature. Other researchers [40,41,42] said that thermal expansion affected the size deviation of the machined surface when milling normalized 42CrMo4 and during orthogonal cutting of AISI 316L Steel. However, the effect of drilling temperature resulting in cutting tool thermal expansion on hole accuracy when drilling CFRP/Ti alloy stacks has not been studied yet. This was a key motivation for providing current research work to address the abovementioned issue.

Concerning the contribution of this paper, we proposed a FEM study of the thermal expansion of the drill and experimental study of the drilling temperature and hole accuracy under various cutting conditions when machining a CFRP/Ti alloy stack. To this aim, the experimental procedure was constructed based on the design of the experiment according to the Taguchi method, consisting of nine trials in which three factors (cutting speed, feed, and time delay) varied on three levels. During the experiment, drilling temperature in the drill and hole accuracy were measured for each hole. The experimental data on drilling temperature was utilized for the calculation of the drill thermal expansion run via FEM in ANSYS software. The results of the hole accuracy measurement combined with drill thermal expansion calculations were used for the identification of hole accuracy deviation. The principal objective of the present work examines the effect of drill thermal expansion on hole accuracy when drilling holes in a CFRP/Ti stack. The results should create favorable conditions for increasing the manufacturing capability of hole machining in CFRP/Ti stacks.

## 2. Materials and Methods

### 2.1. Workpiece Material and Cutting Tool

The multi-material stack used in the current study was composed of two CFRP and titanium alloy plates connected with adhesive. CFRP laminate consisted of 45 unidirectional plies of 0.20 mm thickness, made of IM7 carbon fiber, and Larit (LR285) epoxy resin with the following stacking sequence (0°/90°)_2s_ fabricated using the hand lay-up technique and vacuum bag molding using a vacuum pump in a controlled atmosphere. The total thickness of the CFRP plate is 9 ± 0.01 mm, with 60% fiber volume content. For the definition of carbon fiber volume in the CFRP, nine samples were cut from three CFRP plates. The cut samples were 0.9 cm^3^. They were weighted, and then the epoxy part was burned at a temperature of about 800 °C. The total amount of fiber left after the burning of epoxy was weighed with an accuracy of up to 0.001 g using a Mettler Toledo AB54 Analytical Balance (Mettler-Toledo International Inc., Greifensee, Switzerland). Using multiplication of the fiber weight and density of the chosen fiber, the volume of the fiber in the CFRP sample was calculated. The average value of the carbon fiber for nine samples was defined.

The material of the titanium alloy plate was chosen as Ti-2.5Al-2Mn near α—alloy with a thickness of 8 mm with the following mechanical properties (Table 1). Accordingly, the total thickness of the CFRP/Ti alloy stack was 17 mm. The actual chemical composition of the Ti alloy was defined using a JEOL (JOEL Ltd., Tokyo, Japan) JSM-7600F scanning electron microscope (SEM), equipped with an Oxford Instruments (Oxford Instruments NanoAnalysis & Asylum Research, Abingdon, UK) energy dispersion spectroscopy (EDS) facility. The percentage of the alloying element weight (%) is as follows: Ti—96.42%, Al—1.92%, C—0.21%, O—0.19%, Si—0.17%, Mn—0.89%, Fe—0.20%.

In the present study, nine items of WC9 TiN-TiAlN coated a Ø10 mm twist drill 5510-R-RT100U Guhring (Gühring Slovakia s.r.o., Považská Bystrica, Slovakia) with inner holes for coolant supply have been utilized. Each drill bit was measured using a universal automatic measuring machine for the cutting tool, Zoller Genius 3 s (E. ZOLLER GmbH & Co. KG, Pleidelsheim, Germany). During measurement, it was defined that the geometric parameters of the drill bit were characterized by the actual point angle (2γ = 140.60°), axial relief angle (αa.r = 7.50–8.26°), chisel edge angle (ψ = 45.33–55.62°), helix angle (ω = 29.81–30.10°), drill bit diameter (D, mm), and radial runout of the drill bit (Table 2).

### 2.2. Numerical Setup

For the theoretical study of the research hypothesis of the thermal expansion of the drill bit under various cutting conditions, the finite element model was established. Modeling was established under the finite element method using commercial ANSYS software (ANSYS, Inc., Canonsburg, PA, USA). The basic FE model of the drill was the 3D model of the twist drill 5510-R-RT100U Guhring (Gühring Slovakia s.r.o., Považská Bystrica, Slovakia). In this chapter, the research is focused on the calculation of the linear thermal expansion of the drill. The calculation was carried out with the body temperature linearly increasing for one hundred fifty substeps. The experimental results of the drilling temperature under various cutting conditions were used as input data regarding temperature accumulated in the drill, and as a thermal source on the cutting edge. By assuming that heat input induced by the drilling process tends to produce a homogeneous thermal distribution, it was decided to use a half model with the symmetric condition. A hexahedral eight-node finite element (SOLID70) with one degree of freedom (temperature) was selected for the transient thermal analysis. For the structural analysis, a hexahedral finite element (SOLID185) with three degrees of freedom was used. The FE model consists of 273,979 elements and 410,525 nodes. For the identification of the tool’s thermal expansion under various cutting conditions, nine trials of FE modeling were implemented. In order to calculate the transient temperature field in a drill, Transient Thermal analysis was conducted using ANSYS software (ANSYS, Inc. Canonsburg, PA, USA). When the transient temperature field in the drill was calculated for the duration of the machining time, the thermal deformation of the drill was obtained. The gained results will be used for feature analysis of the thermal expansion influence on hole diameter deviation in the present paper. The heat distribution was calculated in according to Equation (1).
(1)$$q\left(x,y,t\right)\frac{6fQ}{ab\pi }\exp[-\frac{3x^{2}}{b^{2}}-\frac{3y^{2}}{a^{2}}]$$
(2)$$Q=\frac{\rho }{a}\int C_{p}\left(T\right)\cdot (T-T_{0})dV$$
where *x,y*—direction of heat distribution, *t*—time of heat transfer, (s); *f*—front thermal gradient; *Q*-heat flow, (J·mm^−1^); *a*—length of heat input, (mm); *b*—width of heat input, (mm); *ρ*—2013 density, (kg/m^3^); *C_p_*—specific heat, (J/kg·°C); *T*—temperature on the cutting edge, (°C); *T*_0_—environment temperature, (°C); *V*—drill volume, (mm^3^).

#### 2.2.1. Drill Geometry and Boundary Conditions

Cutting tool material WC9 was modeled as a fully isotropic and homogeneous material. The key mechanical properties of WC9 are summarized in Table 3. Schematic illustration of FE model of a drill is shown in Figure 1.

#### 2.2.2. Numerical Results

As a result of FEM, graphs of linear thermal expansion of the drill were obtained under conditions of drilling hole No. 1 for nine trails. Thus, the thermal expansion of the drill was calculated regarding the time instant, which allowed for further synchronizing of the obtained data with the results of the measured accuracy values of the holes in the experimental samples. It was determined that, depending on the drilling temperature, cutting conditions, and machining time, the thermal expansion of the cutting tool varies in the range of 11 to 27 μm at maximum values. Analyzing the thermal expansion values of the drill due to the drilling temperature under various cutting conditions, we can conclude that the maximum thermal expansion of the drill ~ 21 μm was obtained in trails No. 1, No. 4, and No. 7, in which the maximum drilling temperatures of the titanium alloy were 320 °C, 350 °C, and 493 °C (Figure 2a). The machining times were 115 s, 43 s, and 27 s, respectively. At the same time, in experiments No. 2, No. 5, and No. 8, the thermal expansion was ~14 μm, and in tests No. 3, No. 6, No. 9 was ~13 μm. It can be concluded that the increase in the thermal expansion of the drill depends not only on the absolute values of the drilling temperature but also on the machining time (Figure 2b). Since at low values of cutting speed and feed, the machining time is maximum, this creates favorable conditions for heat transition from the cutting zone into the cutting tool. As a result of the FEM calculation, it was found that the thermal expansion of the drill when machining holes in a CFRP/Ti alloy stack with an accuracy of H9 (+36 μm) can be a significant factor affecting the hole accuracy. Further validation of the research hypothesis requires experimental studies that are described in the following sections of the paper.

### 2.3. Experimental Setup

In the present study, the experimental set up was implemented at the 5 axial DMU 85V CNC (Computer Numerical Control) milling center (DMG MORI, Pfronten, Germany) (Figure 3). The CFRP/Ti alloy sample was fixed on a new precise wise Schunk Kontec KSC-F-125 (SCHUNK GmbH & Co. KG, Lauffen/Neckar, Germany) with an error of location of 0.02 mm, which was clamped to the Dynamometer Kistler 9257 (Kistler Group, Winterthur, Switzerland) at the machine table. According to the aim of the study, during drilling trails, the thrust force and drilling temperature were measured. Thrust force was measured using a three-component dynamometer Kistler 9257 (Kistler Group, Winterthur, Switzerland) with a frequency of 200 Hz. The signal of the thrust force was transmitted to a multi-channel amplifier Kistler 5070 (Kistler Group, Winterthur, Switzerland), then to a A/D board and recorded on a personal computer (PC) using DynoWare software. The drilling temperature was measured via a rotating wireless temperature measuring system, which was clamped on an HSK63 collet chuck with a collet screw. The functioning of the system is based on a combination of Seebeck effect measurement with cold junction compensation and wireless high-frequency transmission of the thermocouple signal via Bluetooth channel. This system was powered using Li-ion batteries and consisted of a K-type thermocouple connected to a thermocouple signal amplifier, which transferred the signal to a Bluetooth module. After that, the wireless Bluetooth signal receiver captured the drilling temperature signal and transferred it to the PC with a USB interface converter to the COM port. Self-developed software captured the signal from the COM port and recreated it in real-time on the PC screen and into an Excel file. The thermocouple was embedded under the flank surface of the twist drill in a mechanically-grinded bind groove through the coolant hole at the distance of 1.3 mm from the cutting edge and 1.7 mm from the outer corner. The temperature measurement range of the K-type thermocouple was 40–1036 °C with an accuracy of 0.5 °C. The sampling frequency of the wireless temperature measuring system was 200 Hz, with a signal transmission frequency of 2.4 GHz. Simultaneous measurements of thrust force and drilling temperature made it possible to determine the drill position [45,46,47] and drilling temperature in the hole when drilling the CFRP/Ti stack via a computer time comparison with an accuracy up to 100 ms.

The experimental work included a factorial design L9 (34) Taguchi orthogonal array involving three variations in cutting speed, feed, and time delay in the drilling of adjacent holes with 3 levels (Table 4). The drilling of the CFRP/Ti alloy stack was implemented at a constant and equal cutting speed for both layers of the stack, which varied from 15 m/min to 65 m/min at a feed rate of 0.02–0.08 mm/r. In order to study the effect of heat accumulation in drill and its effect on possible tool expansion, it was proposed to vary the time delay at three levels. The first level involved drilling with so-called “cold drill machining” (CDM), when the drill was cooled to room temperature before drilling the next hole in the trail. The second level was drilling with a 10 s time delay between adjacent holes in the trail. The third level was drilling holes in the trail one by one, so-called “hot drill machining” (HDM), which was drilled with a 5 s time delay between the holes.

The hole accuracy or actual hole diameter was measured on a coordinate measuring machine ZEISS PRISMO ULTRA(Carl Zeiss AG, Oberkochen, Germany). The diameter of the probe was 3 mm. The measurement was conducted at room temperature (20 ± 2 °C) with accuracy up to 1 µm. A total of 24 diameters were measured, acquiring 32 points per each diameter. In such a way, the cloud of 768 measurements was made per each hole.

## 3. Experimental Results and Discussion

### 3.1. Drilling Temperature

The experimental procedure of drilling the CFRP/ Ti alloy stack was implemented according to DOE (Table 3). In each trial, five holes were drilled using a new drill at various cutting parameters. During the measurement of the drill geometry, it was identified that the distance from the chisel edge to the drill’s outer corner was 1.82 mm. On feature graphs, the location of the thermocouple (TC) is markers with a red point. The drilling temperature curves (Figure 4) show its progress through the hole depth when drilling hole No. 1 in various cutting parameters. It was measured that in trail No. 1 (*v* (15); *f* (0.02); *T_d_* (120)), the drilling temperature changes from 58.7 °C at the 0.5 mm depth of the hole, while the steady drilling process of CFRP starts at 70.3 °C at a 6 mm depth, and the heat from cutting the Ti alloy layer had not yet impacted the measured parameter. The drilling temperature of the steady drilling process of the Ti layer at a 10 mm depth of the hole was 225.4 °C and at 15 mm was 320.6 °C. The analysis of drilling temperature curves when drilling hole No. 1 for all trails uncovered that the drilling temperature of the steady process for CFRP machining varied from 48.6 °C to 104.9 °C and in the Ti alloy layer from 189.8 °C to 461.4 °C. The maximal values of the drilling temperature in CFRP were measured during drilling under *v* = 65 m/min and *f* = 0.08 mm/r, while for Ti alloy *v* = 65 m/min and *f* = 0.05 mm/r. Taking into consideration that during the drilling of hole No. 1, the effect of the time delay factor had not yet influenced the drilling temperature; it could be concluded that its value was mostly affected by friction in the tool-workpiece interface zone and the cutting speed, which was reported [48]. Lower cutting speed created more favorable conditions for the transition of heat generated in the cutting zone in workpiece material and chip than in drill.

During machining of hole No. 5 for trails No. 2, No. 3, No. 4, No. 5, No. 7, and No. 9, the effect of the time delay on the drilling temperature was observed (Figure 5). The heat which was accumulated in the tool during drilling of hole No. 1 resulted in drill bit temperatures which varied from 181.7 °C to 251.6 °C. However, drilling temperature in the abovementioned trails started to decrease up to (109.3–208.4 °C) depending on cutting parameters instead of rising when machining CFRP. This trend could be explained by the ability of epoxy resin to dissipate heat levels lower than the glass transition temperature (≈200 °C) [49,50,51]. In such a way, when drilling CFRP, not only heat that was generated but heat that accumulated in the drill after machining of the previous hole was dissipated in CFRP. Though the effect of this phenomenon is limited and at the 7 mm hole depth, the drilling temperature started to become impacted by the thermal energy generated from machining of the Ti alloy layer. Drilling temperature in the Ti layer increases in a range from 182.2 °C to 563.7 °C, which correlated with cutting speed and the effect of heat accumulation when drilling trails No. 2, 3, 4, 5, 7, and No. 9. The combination of high cutting speed and low feed leads to a higher temperature than a combination of high cutting speed and maximal feed. This could be explained by a more significant thickness of the chip under high feed and resulting transition of a larger quantity of heat with a chip, but also by the shorter cutting time during which heat is generated, as reported in Reference [51].

As a result of the analysis, it was found that the drilling temperature is significantly affected by the delay time of the subsequent hole drilling and the machining time. In cutting conditions where drilling was carried out under “cold drill machining” conditions, the drilling temperature in CFRP varied from 75 °C to 125 °C, and in the titanium alloy reached 300–475 °C, which corresponds to the previously described results [7,8,22]. However, in trails close to the actual production conditions with time delay factor *T_d_* (5 to 10 s), the drilling temperature in CFRP varied from 150 °C to 300 °C. Such values significantly exceed the glass transition temperature (≈200 °C) and can negatively affect the quality of the hole wall surface in CFRP. In titanium alloy, the drilling temperature reached 400–625 °C. The effect of the time delay factor on heat balance conditions when drilling CFRP is characterized by a downward curve of the drilling temperature (Figure 5b). It could be concluded that heat dissipation is mostly influenced by the thermomechanical properties of CFRP but not by cutting conditions. This statement is evidenced by the same angle of inclination for the curve of the drilling temperature when drilling CFRP. This knowledge of ours is also in line with the knowledge of the authors [52,53].

### 3.2. Hole Diameter and Roundness

Hole diameter sizes in the CFRP layer varied from 10.009 mm to 11.15 mm while drilling hole No. 1 (Figure 6a). The maximal tolerance field of 0.298 mm between maximal (Ø11.15 mm) and minimal (Ø10.851 mm) dimensions was measured in trail No. 9 (*v* (65); *f* (0.08); *T_d_* (10)). During the analysis of hole diameter curves, it was noticed that maximal deviations were measured in the holes of trail No. 1, No. 3, No. 4, and No. 9. This could be explained because of the sliding of the rough Ti alloy chip over the hall wall generated under the high feed of 0.08 mm/r for trials No. 3 and No. 9 and the relatively long machining time as a result of the combination of low cutting speed and low feed for trails No. 1 and No. 4, which corresponded with Reference [22]. In the Ti alloy layer, the actual hole size varied from 10.028 mm to 10.081 mm. A maximal tolerance field of 0.021 mm between the maximal (Ø10.057 mm) and minimal (Ø10.036 mm) dimensions was measured in trail No. 8 (*v* (65); *f* (0.05); *T_d_* (120)) (Figure 6a).

According to Figure 6b, it is supposed that the increase in hole diameter was influenced by the sliding of the rough Ti alloy chip at the CFRP hole wall generated under the high feed rate as it was described in [22,23,26]. At the same time, in trail No. 6, which was also run under high feed but also under CDM conditions, the deviation of hole diameters varied from 10.070 mm to 10.127 mm. The factors which differ with the three abovementioned trails, except cutting speed, were HDM and CDM conditions. So, it could be concluded that the shrinkage of the hole diameter in CFRP was affected not only by chip sliding but also by the drill bit temperature. In the Ti alloy, the hole diameter varied from 10.034 mm (trail No. 6) to 10.120 mm (trail No. 7). Deviation of the hole diameter in the Ti alloy was affected by the radial run out of the drill (Table 2), with feed per minute as a function of time being necessary for the positioning of the chisel edge, similar to what was observed in Reference [28], and drill bit temperature. It was observed that for trail pairs No. 2 (*v_f_* = 23.85 mm/min) and No. 4 (*v_f_* = 25.46 mm/min), No. 3 (*v_f_* = 38.16 mm/min) and No. 7 (*v_f_* = 41.38 mm/min), and No. 6 (*v_f_* =101.84 mm/min) and No. 8 (*v_f_* = 103.45 mm/min), the hole diameter and its deviations were bigger in trails No. 4, No. 7, and No. 8, where the drill bit temperature was higher in comparison to the drilling temperature in trails No. 2, No. 3, and No. 6. It appears to contrast to the findings in Reference [8] that the hole diameter oversize mostly rises with a feed rate increase. In Reference [54], it was reported that the deviation of hole diameter could be reduced up to 60 µm in CFRP and Ti alloy layers by utilizing peck drilling techniques in combination with the MQL technique. It creates favorable conditions for chip crushing and minimizes heat generation in the interface tool-workpiece zone. In an experimental study on drilling mechanisms and strategies of CFRP/Ti stacks [27] which run under the same range of cutting parameters and almost similar tool geometry, authors reported for CFRP-Ti a cutting about up to about 175 µm of deviation in the hole diameter in CFRP and up to 25 µm in the Ti alloy, which totally corresponds to the results reported in the present study. However, the components of which the hole diameter deviation consisted were not reported. Based on the measurement of drill bit geometry (Table 2), results of FEA of drill bit thermal expansion (Figure 2), and measurements of hole size (Figure 6), the components of hole diameter deviation were evaluated (Figure 7).

The error of drill geometry consists of a deviation of drill bit diameters and radial runout. The thermal expansion of the drill is presented as a sum of the actual values of thermal expansion, gained via FEA, and the error of the drill geometry. It should be specified that the effect of the drill thermal expansion in CFRP is minimal and could be observed properly because of the tremendous effect of sliding the chip over the hole wall surface. However, taking into consideration that the deviation of hole diameter in the Ti alloy varied from 28 µm to 81 µm, the values of thermal expansion and error of drill geometry have significant effects. The average percentage of thermal expansion in the hole diameter deviation in Ti alloy varied from 15% to 37% with respect to cutting conditions and drilling temperature. The average percentage of error of drill bit geometry varied from 12% to 37%. In the reverse scenario, the sum of these two components provides up to 74% of the hole diameter deviation in Ti alloy. The rest is probably influenced by changeable dynamic conditions in the interface zone of the tool-workpiece, vibrations, and inhomogeneous hardness of the alloy. In References [16,17,18,19] the effect of tool wear on hole diameter was reported. Figure 7 takes the discussed results into consideration. In the drilling of the first hole in the trails, the effect of tool wear could not be observed.

Feature analysis of hole roundness uncovers trends for a significant effect of cutting conditions and drilling temperature on hole roundness in CFRP (Figure 7). In References [27,28,36], roundness varied from 15 µm to 35 µm, which correlates with the results. Taking into consideration the measurement of the drilling temperature (Figure 4b) and the related discussion, it becomes more evident that the distribution of heat in CFRP reduces the adhesion connections between the fiber and epoxy. In research focused on studying frictional heat and cutting temperature of CFRP [48] as well as in the study of temperature-dependent properties of CFRP, the phenomenon of decreasing of interfacial shear strength and fracture toughness was reported [38]. In Reference [22] it was described as the indirect connection of drilling temperature and residual stress under the machined surface of the CFRP/Ti alloy. However, the results regarding the actual effect of the reduction of CFRP mechanical properties on hole diameter or roundness have not yet been produced. In the present study, the analysis of roundness is presented (Figure 8).

The comparison of results of hole roundness for holes No. 1 and No. 5 concerning cutting conditions showed that when machining under CDM conditions (Figure 8a,c), the distributions of measured roundness are equal for hole No. 1 and hole No. 5, respectively. However, in trials in which holes were machined under HDM conditions (Figure 8b,d), there is a significant distribution of roundness between hole No. 1 and when the time delay factor has not taken effect yet and hole No. 5 influenced by time delay factor. Under HDM conditions, hole roundness in CFRP increases from 75 µm to 120 µm concerning cutting conditions and machining time. Such a phenomenon is explained by the decrease of the thermal-dependent properties of CFRP. In Ti alloy, such an effect was not observed.

## 4. Conclusions

The current work investigates the influence of cutting parameters (cutting speed, feed rate) on the drilling temperature and hole accuracy when drilling the CFRP/Ti alloy stack under various time delay factors. During experimental research, the drilling temperature was measured in the rotating drill utilizing a K-type thermocouple sensor embedded under the flank surface of the drill and connected to the wireless device. The effect of the drilling temperature on drill thermal expansion was studied using finite element analysis. Hole accuracy was measured using a CNC coordinate measuring machine. The following results can be concluded:The experimental study shows that the time delay factor has a significant influence on drilling temperature, which increased from 49% to 62% in CFRP and 14% to 29% in Ti alloy with respect to cutting parameters. At the same time, in trails that were run without the effect of time delay under “cold drill machining” conditions, the drilling temperature increased from 13% to 32% in CFRP and 10% to 27% in Ti alloy.During the measurement of drilling temperature when drilling the CFRP/Ti alloy stack under 5 s and 10 s time delay, the heat dissipation effect in CFRP was noticed, which was characterized by decreasing drilling temperatures.Heat dissipation in CFRP reduced the thermal-dependent properties of CFRP, which resulted in hole roundness increases in the range of 75 µm to 120 µm corresponding to cutting conditions and machining time.It was found that the hole taper in CFRP is influenced by the combination of the Ti alloy chip sliding effect and the thermal expansion of the tool. At the same time, in Ti alloy, the hole taper is influenced by the combination of the radial run out of the drill and thermal expansion of the tool. Based on the measurement of hole diameter with the CNC coordinate measuring machine combined with the results of drilling temperature values, it was determined that increasing drilling temperatures up to ≈ 600 °C leads to the rising of the hole diameter deviation in Ti alloy up to 56 µm. In other trails, where drilling temperature was in the range of (350–500 °C), the deviation of hole diameter varied from 8 µm to 22 µm.The maximal thermal expansion of the drill varied from 11 µm to 27 µm and impacted from 12% to 37% of the total deviation of the hole diameter in Ti alloy corresponding to cutting conditions.Wireless measurement of the drilling temperature used in the current study suggested the possibility of real-time temperature monitoring to study the wear rate of the drill, the drill geometry’s effect on hole diameter, and roundness when drilling. The phenomenon of thermal expansion reported in the present study can be used as an input for future studies for the optimization of cutting conditions when drilling CFRP/Ti alloy stacks.

The influence of tool wear on the accuracy of a drilled hole is very significant. This is especially true for wear on the tips of cutting wedges, the so-called dimensional wear. The authors did not address this wear.

In the future, several field experiments are planned in order to assess the adequacy of the proposed experimental procedure and the use of FEM software. The aim will be to extend the research to various other materials [55], to other machining methods [56], to other types of drilling tools, to setting up the experiment [57] with the ambition of predicting various parameters as a result of drilling [58], and to optimizing technological conditions [59]. These may be experiments in the technological laboratory of the Center of Excellence for 5-Axis Machining at the Slovak University of Technology in Trnava.

## Figures and Tables

**Figure 1 materials-13-03232-f001:**
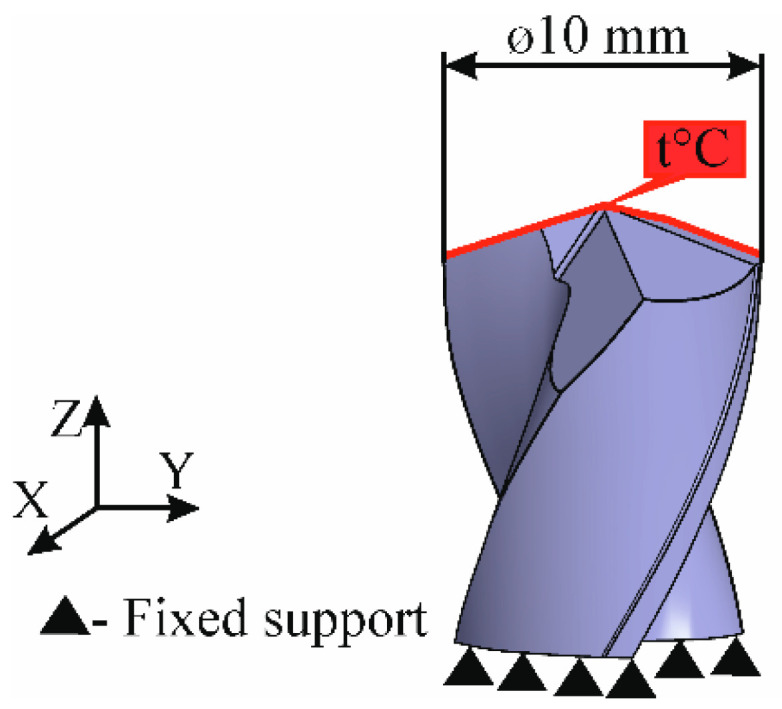
Schematic illustration of the FE model of a drill.

**Figure 2 materials-13-03232-f002:**
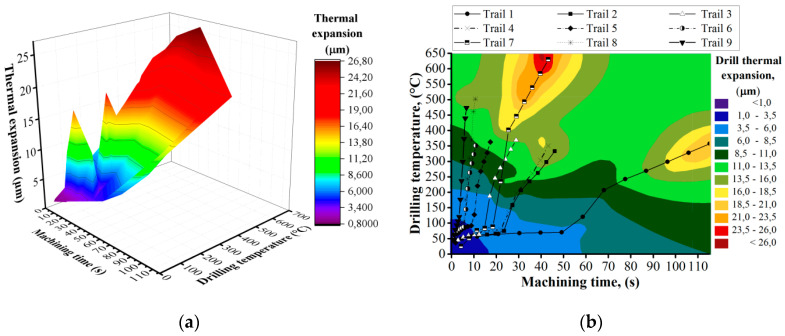
Results of finite element modeling of drill thermal expansion. (**a**) evaluation of drilling thermal expansion depending on machining time and drilling temperature; (**b**) Drill thermal expansion Plane Projection.

**Figure 3 materials-13-03232-f003:**
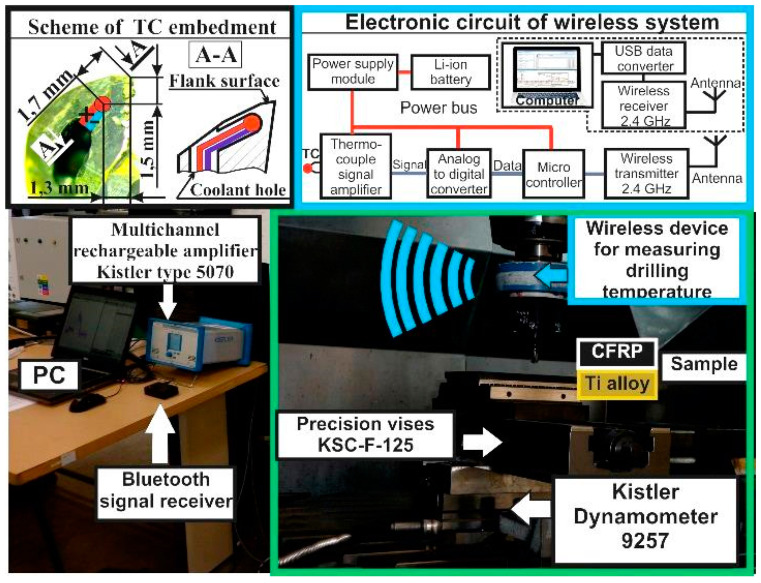
Experimental setup of the drilling tests.

**Figure 4 materials-13-03232-f004:**
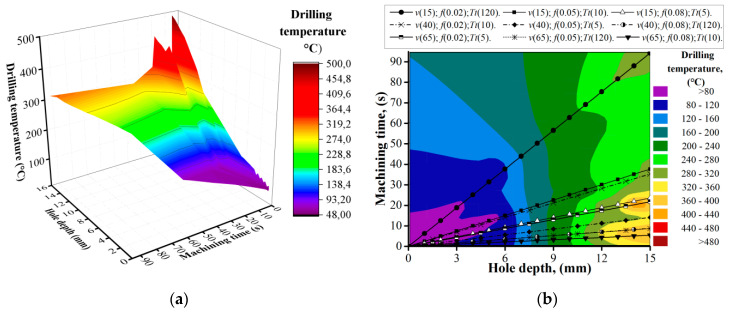
Drilling temperature when machining hole No. 1 in CFRP/Ti alloy stack; (**a**) evaluation of drilling temperature depending on machining time and hole depth; (**b**) Drilling Temperature Plane Projection.

**Figure 5 materials-13-03232-f005:**
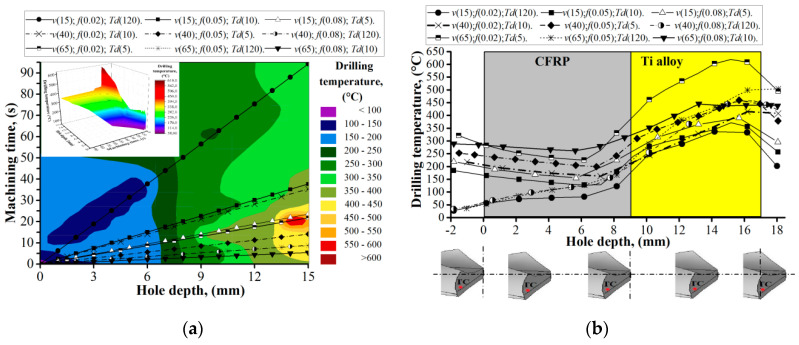
Drilling temperature when machining hole No. 5 in the CFRP/Ti alloy stack. (**a**) Distribution of the drilling temperature with respect to machining time and hole depth; (**b**) drilling temperature evaluation of the hole depth.

**Figure 6 materials-13-03232-f006:**
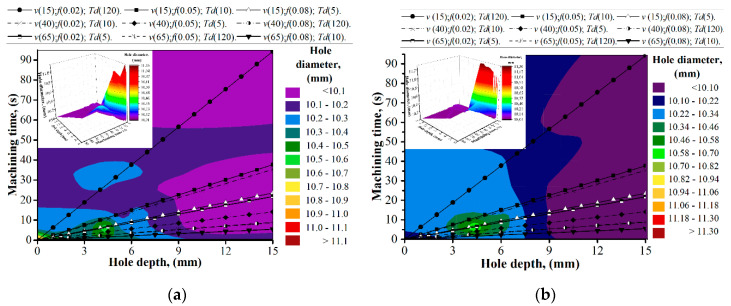
The hole diameter measures in hole No. 1. (**a**) Distribution of hole diameter under the conditions of the machining time and hole depth for hole No. 1; (**b**) distribution of the hole diameter in terms of machining time and hole depth for hole No. 5.

**Figure 7 materials-13-03232-f007:**
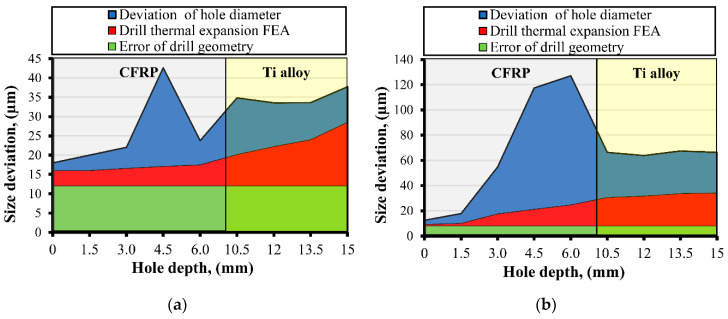
Hole diameter measured in hole No. 1. (**a**) Components of hole diameter deviation for hole No. 1 trail No. 2 (*v* (15); *f* (0.05); *T_d_* (120)); (**b**) components of hole diameter deviation for hole No. 1 trail No. 7 (*v* (65); *f* (0.02); *T_d_* (5)).

**Figure 8 materials-13-03232-f008:**
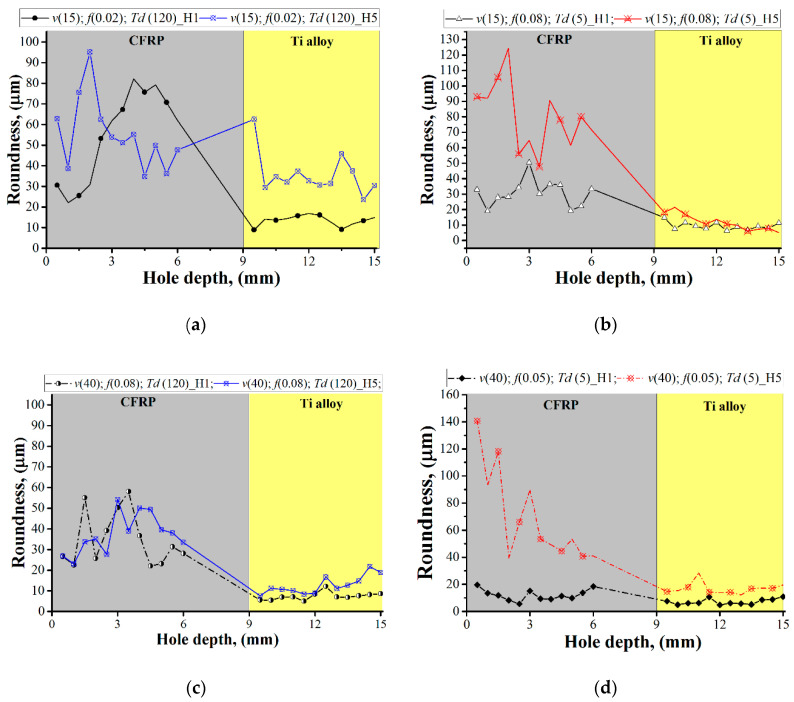
The effect of the time delay factor on hole roundness. (**a**,**c**) Comparison of hole roundness for holes No. 1 and No. 5 drilled under CDM conditions; (**b**,**d**) comparison of hole roundness for holes No. 1 and No. 5 drilled under HDM conditions.

**Table 1 materials-13-03232-t001:** Mechanical properties of the Ti-2.5Al-2Mn alloy [43].

Tensile Strength (MPa)	Modulus of Elasticity (GPa)	Density (kg/m^3^)	Elongation (%)	Thermal Conductivity (W/m K)	Hardness (HV)
735	115	4550	10	9.63	178

**Table 2 materials-13-03232-t002:** Drill bit geometry.

Geometric Parameters	Drill Number in Respect to Trail Number								
	1	2	3	4	5	6	7	8	9
*D* (mm)	10.008	10.003	10.003	10.003	10.000	10.000	10.000	10.000	10.000
Radial runout (mm)	0.010	0.012	0.016	0.008	0.008	0.008	0.008	0.008	0.008

**Table 3 materials-13-03232-t003:** Mechanical properties of drill tool materials (WC9) [44].

**Density (kg/m^3^)**	**Coefficient of Thermal Expansion (°C^−1^)**	**Zero-Thermal Expansion (°C)**	**Young’s Modulus (GPa)**	**Poisson’s Ration**	**Bulk Modulus (GPa)**	**Shear Modulus (GPa)**
15700	4.8	22	615	0.24	643	274
**Compressive Yield Strength (MPa)**	**Tensile Ultimate Strength (MPa)**	**Isotropic Thermal Conductivity (W/mm·°C)**			**Specific Heat (J/kg·°C)**	**-**
		**177 °C**	**427 °C**	**876 °C**		
4780	350	0.38	0.44	0.52	434	-

**Table 4 materials-13-03232-t004:** Drilling performances and their levels and the design of the experiment according to the Taguchi orthogonal array L_9_.

Drilling Performance		Levels of Factors			Coding DOE in According to Taguchi L9								
					Trail Number								
		1	2	3	1	2	3	4	5	6	7	8	9
**A**	Cutting speed *v* (m/min)	15	40	65	1A	1A	1A	2A	2A	2A	3A	3A	3A
**B**	Feed, *f* (mm)	0.02	0.05	0.08	1B	2B	3B	1B	2B	3B	1B	2B	3B
**C**	Time delay *T_d_* (s)	CDM (120)	10	HDM (5)	1C	2C	3C	2C	3C	1C	3C	1C	2C

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
