# Peer review of "Experimental Study of Drilling Temperature, Geometrical Errors and Thermal Expansion of Drill on Hole Accuracy When Drilling CFRP/Ti Alloy Stacks"

_materials, 2020, doi:10.3390/ma13143232_

Round 1
Reviewer 1 Report
The paper reports study of effect of drilling parameters on hole accuracy and roundness. This study is composed of experimental an numerical parts. Experimental part is focused on the influence of drilling parameters and temperature on the resulting, measured after test, hole quality. Numerical part, for unknown reason, is focused on thermal expansion of the drill. By my understanding, since aim of this study is to investigate final hole accuracy, it shall be focused on behaviour of drilled material (CFRP/Ti composite). In the case of such sophisticated material exhibiting non-homogeneous response to thermo-mechanical loads, coupled thermo-mechanical problem have to be solved to understand investigated phenomena (residual stress, deformation, damage etc.). Without this part presented study is incomplete and, as the authors mentioned "question of ensuring hole quality" still will be "challenging task".
Author Response
Response to Reviewer 1 Comments
Comment: The paper reports study of effect of drilling parameters on hole accuracy and roundness. This study is composed of experimental an numerical parts. Experimental part is focused on the influence of drilling parameters and temperature on the resulting, measured after test, hole quality. Numerical part, for unknown reason, is focused on thermal expansion of the drill. By my understanding, since aim of this study is to investigate final hole accuracy, it shall be focused on behaviour of drilled material (CFRP/Ti composite). In the case of such sophisticated material exhibiting non-homogeneous response to thermo-mechanical loads, coupled thermo-mechanical problem have to be solved to understand investigated phenomena (residual stress, deformation, damage etc.). Without this part presented study is incomplete and, as the authors mentioned "question of ensuring hole quality" still will be "challenging task".
Response: Yes, one can agree with the opponent's opinion that the CFRP / Ti material is very sophisticated because it shows inhomogeneity in each of its volumes and it is not easy to solve the thermomechanical load in the machining process, e.g. as in our case in the drilling process. The authors tried to make a connection between the experimental part and the numerical part in such a way that the experimental results of the drilling temperature under different cutting conditions were used as input data on the temperature accumulated in the drill and as a heat source at the cutting edge. An attempt was made to use the results in a characteristic analysis of the effect of thermal expansion on the deviation of the hole diameter in this document. The results of FEM calculation of thermal expansion of drill and hole accuracy are compared and assumed in graphs Figure 7. a) and Figure 7. b). In the future, it is planned to lead research towards a deeper connection between the experimental and numerical parts.
Reviewer 2 Report
This is an interesting study on the relationship of different parameters that coexist during the drilling of holes in CFRP/Ti alloy stacks.
There are some aspects that need to be improved:
Line 135; the numbers 2 accompanying the direction of the fibres should be subscripts and not superscripts.
Line 136; the percentage by volume of fiber how has it been obtained?
Table 1; How were the values in this table obtained?
Table 3; How were the values in this table obtained?
Figure 3; Everything is too small and concentrated. It is very difficult to see anything. It should be enlarged and perhaps divided into two or three figures.
Author Response
Response to Reviewer 2 Comments
Comment:
This is an interesting study on the relationship of different parameters that coexist during the drilling of holes in CFRP/Ti alloy stacks. There are some aspects that need to be improved:
Point 1: Line 135; the numbers 2 accompanying the direction of the fibres should be subscripts and not superscripts.
Response 1: Yes, we agree with the opponent. Description of fiber direction was corrected in according to opponent comment. Correction from [02/902] to [0°/90°]2s .
Point 2: Line 136; the percentage by volume of fiber how has it been obtained?
Response 2: For explanation, we have inserted the following text:
“Inserted: For definition of carbon fiber volume in the CFRP, nine samples were cut from three CFRP plates. Size of cut samples was of 0.9 cm3. They were weighted. Then the epoxy part was burned in the own at a temperature about 800°C. Total amount of fiber which left after burning of epoxy was weighted with accuracy up to 0.001 g at Mettler Toledo AB54 Analytical Balance. By multiplication of fiber weight and density of chosen fiber the volume of fiber in CFRP sample was calculated. The average value of carbon fiber for nine samples was defined.”
Point 3: Table 1; How were the values in this table obtained?
Response 3: The values in Table 1. were obtained from reference [43].
Point 4: Table 3; How were the values in this table obtained?
Response 4: The values in Table 3. were obtained from reference [44].
Point 5: Figure 3; Everything is too small and concentrated. It is very difficult to see anything. It should be enlarged and perhaps divided into two or three figures.
Response 5: Yes, we agree with the opponent's opinion. We decided to enlarge the Figure 3. to the full width of the page.
Reviewer 3 Report
The paper is interesting with a correct approach. Formally it needs some improvement. Examples:
-line 35 incorrect use of the sign ;
-line 143 the table is split
-line 179 the description of table is split from table.
-fig. 3 can be improved
Author Response
Response to Reviewer 3 Comments
Comment:
The paper is interesting with a correct approach. Formally it needs some improvement. Examples:
Point 1: line 35 incorrect use of the sign ;
Response 1: We assume that the opponent means Ra. Yes, opponent is right, Ra is not a surface roughness but a surface roughness parameter. Therefore, we have deleted the designation Ra.
Point 2: line 143 the table is split
Response 2: Yes, the opponent is right, table 1 is split. This was because it is located at the end of the page, so we moved Table 1. to the next page.
Point 3: line 179 the description of table is split from table.
Response 3: Yes, the opponent is right, description of table 1 is split from table. This was because it is located at the end of the page, so we moved description of Table 1. to the next page.
Point 4: fig. 3 can be improved
Response 4: Yes, we agree with the opponent's opinion. We decided to enlarge the Figure 3. to the full width of the page.
Reviewer 4 Report
Direct measurement of the cutting edge temperature of the WC9 TiN-TiAlN coated Ø10 mm twist drill was carried out in the laboratory. The main advantage of the article is the realizung of this fine physical experiment. The dramaturgy of the article is extremely complex. The first part 2.2 is devoted to the FEM calculation of the thermal expansion of a cutting tool from WC9 without coating during cutting of bilayer CFRP / Ti alloy composite. In this case, the influence of the thermal conductivity of the coating, its triboxidation and wear is not taken into account, which reduces the importance of this section to explain the experimental data. In this section, it is necessary to indicate which later obtained experimental data were used in the calculations. Otherwise, this section is very indirectly related to the article.
In part 2.2 of the article, it is necessary to indicate the literary source of information on the physical properties of WC9 (Table 3), including Coefficient of Thermal Expansion. Obviously this was a book by Alexey S. Kurlov Aleksandr I. Gusev DOI 10.1007 / 978-3-319-00524-9. Reference to the source must be specified. It is necessary to present the temperature dependencies in FEM calculations. Obviously, in the model, during cutting the temperature varied linearly with time. In this case, add the phrase: The calculation was carried out with the body temperature linearly increasing for one hundred (for ex.) substeps. How this was taken into account in the program? If the experimentally obtained data were used in the model, this must be indicated. In fact, the temperature increases nonlinearly depending on the cutting time.
Analytical and FEM calculations stress, strain –strengthening, temperature distributions in the edge of cutting tools were carried out in [GSFox-Rabinovich, G. Totten, Self-Organization during friction, Advanced Surface-Engineered Materials and Systems Design, Tailor & Francis , 2007] confirm this. I recommend adding this publication to the choice of authors, this will strengthen this article. By the way, this is shown in Figure 5-b. With increasing temperature, the cutting tool strength decreases, its wear increases, which also affects the change in hole diameter and accuracy. Unfortunately, this factor is not discussed in the article.
3 Experimental part is executed and presented in the article is excellent.
Many factors affect on drilling accuracy. For this reason, it is very difficult to foregone conclusion that the delay time or thermal expansion of the tool is decisive factors. In my opinion, tool wear is the most significant factor that is not discussed in the article. I advise the authors to supplement this reason in the discussion of the results. We can discuss about interpretation of the results, but an experienced reader can extract a lot of useful information from the presented experimental results.
Author Response
Response to Reviewer 4 Comments
Point 1: Direct measurement of the cutting edge temperature of the WC9 TiN-TiAlN coated Ø10 mm twist drill was carried out in the laboratory. The main advantage of the article is the realizung of this fine physical experiment. The dramaturgy of the article is extremely complex. The first part 2.2 is devoted to the FEM calculation of the thermal expansion of a cutting tool from WC9 without coating during cutting of bilayer CFRP / Ti alloy composite. In this case, the influence of the thermal conductivity of the coating, its triboxidation and wear is not taken into account, which reduces the importance of this section to explain the experimental data. In this section, it is necessary to indicate which later obtained experimental data were used in the calculations. Otherwise, this section is very indirectly related to the article.
Response 1: Yes, we agree with the opponent's comment. The authors tried to make a connection between the experimental part and the numerical part in such a way that the experimental results of the drilling temperature under different cutting conditions were used as input data on the temperature accumulated in the drill and as a heat source at the cutting edge. An attempt was made to use the results in a characteristic analysis of the effect of thermal expansion on the deviation of the hole diameter in this document. In the future, it is planned to lead research towards a deeper connection between the experimental and numerical parts.
Point 2: In part 2.2 of the article, it is necessary to indicate the literary source of information on the physical properties of WC9 (Table 3), including Coefficient of Thermal Expansion. Obviously this was a book by Alexey S. Kurlov Aleksandr I. Gusev DOI 10.1007 / 978-3-319-00524-9. Reference to the source must be specified. It is necessary to present the temperature dependencies in FEM calculations. Obviously, in the model, during cutting the temperature varied linearly with time. In this case, add the phrase: The calculation was carried out with the body temperature linearly increasing for one hundred (for ex.) substeps. How this was taken into account in the program? If the experimentally obtained data were used in the model, this must be indicated. In fact, the temperature increases nonlinearly depending on the cutting time.
Response 2: We agree with the opponent and we have added a literary source [44] in Table 3. and we added the text for explanation: “The calculation was carried out with the body temperature linearly increasing for one hundred fifty substeps.” Line 162/163.
Point 3: Analytical and FEM calculations stress, strain –strengthening, temperature distributions in the edge of cutting tools were carried out in [GSFox-Rabinovich, G. Totten, Self-Organization during friction, Advanced Surface-Engineered Materials and Systems Design, Tailor & Francis , 2007] confirm this. I recommend adding this publication to the choice of authors, this will strengthen this article. By the way, this is shown in Figure 5-b. With increasing temperature, the cutting tool strength decreases, its wear increases, which also affects the change in hole diameter and accuracy. Unfortunately, this factor is not discussed in the article.
Response 3: Yes, we agree with the opponent's opinion and we have added the mentioned literary source [52,53] and a link to the text: "This knowledge of ours is also in line with the knowledge of the authors [52,53]" - line 304/305.
Round 2
Reviewer 1 Report
According to authors response, study was intentionally focused on influence of drill temperature on the hole accuracy. If such limited approach is still of readers interest, I can recommend it for publishing. For the future I would suggest to extend numerical analysis and include effects taking place in the drilled material.